# Characterization of Stem Nodes Associated with Carbon Partitioning in Maize in Response to Nitrogen Availability

**DOI:** 10.3390/ijms23084389

**Published:** 2022-04-15

**Authors:** Yujing Zhao, Peng Ning, Xiaojie Feng, Hanshuo Ren, Mingtang Cui, Lu Yang

**Affiliations:** 1Key Laboratory of Plant-Soil Interactions (Ministry of Education), College of Resources and Environmental Sciences, National Academy of Agriculture Green Development, China Agricultural University, Beijing 100193, China; 13466897383@163.com (Y.Z.); ningpeng2006@163.com (P.N.); 18810760157@163.com (X.F.); rhs42195@163.com (H.R.); 15753228987@163.com (M.C.); 2Key Laboratory of Biology and Genetic Improvement of Oil Crops, Ministry of Agricultural and Rural Affairs, Oil Crops Research Institute of the Chinese Academy of Agricultural Sciences, Wuhan 430062, China

**Keywords:** carbon partitioning, nitrogen, stem node, sugar transporters, vascular bundles

## Abstract

Stem node has been found to be a hub for controlling mineral nutrient distribution in gramineous plants. However, the characteristics of stem nodes associated with whole-plant carbon partitioning in maize (*Zea mays* L.) and their responses to nitrogen (N) availability remains elusive. Maize plants were grown in greenhouse under low to high N supply. Plant growth, sugar accumulation, and sugar transporters in nodes and leaves, as well as the anatomical structure of nodes, were investigated at vegetative phase. When compared to N-sufficient plants, low-N availability stunted growth and resulted in 49–64% less sugars in leaves, which was attributed to low photosynthesis or the accelerated carbon export, as evidenced by more ^13^C detected further below leaf tips. Invariably higher sugar concentrations were found in the stem nodes, rather than in the leaves across N treatments, indicating a crucial role of nodes in facilitating whole-plant carbon partitioning. More and smaller vascular bundles and phloem were observed in stem nodes of N-deficient plants, while higher sugar levels were found in the bottom nodes than in the upper ones. Low-N availability upregulated the gene expressions of sugar transporters, which putatively function in nodes such as *ZmSWEETs* and *ZmSUTs* at the bottom stem, but suppressed them in the upper ones, showing a developmental impact on node function. Further, greater activity of sugar transporters in the bottom nodes was associated with less sugars in leaves. Overall, these results highlighted that stem nodes may play an important role in facilitating long-distance sugar transport in maize.

## 1. Introduction

Whole-plant carbohydrate partitioning underlines all aspects of plant growth and crop yield. It is the process whereby photoassimilate is distributed from the source leaves through the veins to the non-photosynthetic tissues, such as roots, stems, flowers, fruits, and seeds [1,2]. In addition to the carbon consumed by shoot growth and grain yield formation, approximately 20 to 60% of net photosynthetically fixed carbon is allocated belowground in annual plants to support root growth, rhizodeposition, and feeding microbes [3,4]. This whole-plant carbon partitioning is involved in multiple processes of sugar transport from the sources to the sinks, which is under genetic control, hormone regulation, and also susceptible to changing environments, such as nutrient availability [5,6,7]. Thus, a better understanding of the mechanisms through which carbohydrate is allocated in plants is needed to facilitate crop improvement [1,8].

Carbon distribution in plants entails sugar accumulation and export in the source leaves, long-distance transport in the vascular bundles, and utilization in the sink tissues [8,9]. Carbohydrate is transported primarily in the form of sucrose through phloem in most higher plant species [2]. In maize and other cereals, sucrose is mainly transported to immature leaves and developing roots during the vegetative stage, and to developing seeds during the reproductive stage. Stem node is a central location connecting stem, leaves, and axillary buds/tillers in Poaceae, and there are highly developed and fully organized vascular bundles in this region [10,11]. Studies have found that in rice and barley, node acts as a hub to control the allocation of mineral elements e.g., phosphorus, zinc, and silicon to the vegetative organs and grains, which are mainly mediated by the node-based transporters [11,12,13]. Although minerals and sugars are translocated through different routes i.e., xylem vs. phloem, they are both concentrated in the nodal region to facilitate their distribution to other tissues. Hence, it is assumed that nodes are crucial players in the whole-plant carbon partitioning in maize. However, it remains elusive that how stem nodes act in the whole-plant carbon partitioning in maize.

Each axial vascular bundle in the stem spans across several nodes in Poaceae, and have specialized as diffuse, intermediate, and enlarged vascular bundles, which are involved in active inter-vascular transfer processes of solutes [11]. Maize plants adopt an apoplasmic strategy for sucrose loading and unloading [1,14], which are mediated by sugar transporters, such as sucrose transporters (SUTs), or sugars will eventually be exported by transporters (SWEETs) [14,15,16,17,18]. Studies have shown that *ZmSUT1* is expressed in multiple cell types throughout the maize plant, and functions in phloem loading and retrieval of leaked sucrose from apoplasm. Its expression is broad initially in the developing stem but becomes restricted in mature stem veins [15]. Even at 10 days after pollination, *ZmSUT1* has high expression activity in the internode [19]. It is speculated that, in the nodes, *ZmSUT1* also mediates uptake of sucrose from the apoplasm. Although uncharacterized, SWEETs putatively play roles in facilitating the release of sucrose into the sieve element apoplasm [20]. They may also promote facilitated diffusion of sucrose into the apoplasm of mature stems in cereals [20,21]. It is interesting to characterize the expression activity of these transporters in stem nodes regarding sugar translocation and whole-plant carbon partitioning in maize.

Nitrogen (N) is one of the critical nutrients affecting plant growth, and it has been well documented that N availability regulates carbon fixation and export and alters the partitioning of the fixed carbon in plants [6,22,23]. There is a key debate as to whether carbon partitioning under low N availability is source- or sink-limited, i.e., carbon assimilation or assimilate usage, respectively. Considerable evidence is accumulating that source and sink factors co-limit this process [24,25]. In addition to the source–sink interactions, N availability affects carbon fluxes in vascular bundles via altering the structure of vascular bundles such as phloem number and size. Alternatively, this can be implemented through the function of sugar transporters involved in sucrose export, loading, and unloading in phloem [6,26]. It has been found that N supply significantly increases the number of vascular bundles in the peduncle and cob tissues of maize, which facilitates grain filling [26]. Further, lower N availability down-regulated sucrose transporters of *ZmSWEET13a*, *ZmSWEET13c*, and *ZmSUT1* in maize leaves [6], suggesting a limited function of sucrose phloem loading under low N conditions. However, it remains elusive as to how maize stem nodes respond to N availability in relation to carbon export from leaves and distribution to other tissues.

To address some of these questions, we conducted a greenhouse experiment to characterize maize stem nodes associated with carbon partitioning in response to N application (low, medium, and high N levels). Analogous to the mediation function of stem nodes in mineral nutrients’ allocation in other cereals, we hypothesized that, being a junction of maize plant, the stem node plays a crucial role in sugar export from leaves and translocation to other tissues, which is accomplished through the anatomical changes of the vascular bundle and expression activities of sugar transporters under different N availability.

## 2. Results

### 2.1. Maize Growth, N Uptake, and Photosynthesis

At 62 days after silking (DAS), when compared with MN and HN, low N availability (LN) significantly reduced shoot dry matter by 63–65%, root dry weight by 65–66%, and total N uptake by 81–84% in maize (Table 1; Figure 1). Nitrogen level had limited impacts on root-to-shoot dry weight ratio at this stage. As expected, the relative chlorophyll content and net photosynthesis of leaves in LN plants were 63–66% lower than in the N-sufficient plants. Except for total N uptake, there were no statistical differences detected for the above parameters between the MN and HN treatments (Table 1). In addition, the grain yield of the N-deficient plants was only 10–14% of the MN and HN plants (Figure 1; Appendix A).

### 2.2. Soluble Sugar Concentration of Leaf and Nodes

In comparison with the MN and HN, the concentrations of soluble sugars (sum of fructose, glucose, and sucrose) were invariably lower in the eighth leaf and sampled nodes of low-N plants, with a few exceptions in Node 3 (Appendix A). Low-N significantly decreased glucose by 49–52%, fructose by 63–64%, and sucrose by 59–63% in the eighth leaf when compared to the MN and HN treatments (Figure 2). Similar patterns were observed for glucose in the stem nodes in response to N availability, however, sucrose concentrations were less affected by N rates. The differences in fructose concentration were mainly found between the LN and MN treatments (Figure 3). Furthermore, sugar concentrations were higher in the nodes than in the leaves, as well as more sugars in the stem base node than in the upper ones (Figure 2 and Figure 3).

### 2.3. ^13^C-Sucrose Export in Leaves

Overall, the total amount of ^13^C detected in the labeled leaf fragments was comparable between N treatments, but exhibited different patterns of export and distribution (Figure 4). Low N availability caused 25–26% reductions of the amount of ^13^C remained at the application fragment when compared to the MN and HN (*p* < 0.05), while more ^13^C was detected at 0–2 and 2–4 cm below the labeling section. In the leaf fragments at 4–6 and 6–8 cm, a considerable amount of ^13^C was exported in the low-N leaves, but rarely detected in the MN and HN leaves (Figure 4A). Regarding the proportions of ^13^C distribution, a similar pattern was observed with the ^13^C amount in different leaf fragments between N treatments (Figure 4B).

### 2.4. Anatomic Analysis of Stem Nodes

When compared to the LN plants, significantly larger but less vascular bundles and phloem were observed in all examined nodes under MN treatment (Figure 5 and Figure 6). On average across node position, low N reduced vascular bundle size by 78% and phloem size by 84% more than the MN plants, while it increased the number of vascular bundles per unit area by 36% (Figure 6). Besides that, similar patterns were observed in the internodes, and less vascular bundles in the internodes were also found in the LN vs. MN plants (Appendix A).

### 2.5. Transcript Abundance of Genes Involving in Sugar Transport in Leaves and Nodes

Genes encoding ZmSWEETs (*ZmSWEET3a*, *ZmSWEET4a*, *ZmSWEET4b*, *ZmSWEET13a*, *ZmSWEET13b*, and *ZmSWEET13c*), and ZmSUTs (*ZmSUT1*, *ZmSUT2* and *ZmSUT4*) were putatively characterized, as they are important for sugar transport in maize plants [14,15,17,20,21]. The expression profiles of these genes in maize leaves (Figure 7) and stem nodes (Figure 8) were compared under different N availability. All *ZmSWEET* and *ZmSUT* genes examined in the present study were significantly up-regulated by low N availability in Node 1, Node 2, Node 3, and the eighth leaf (with a few exceptions in the eighth leaf). On the contrary, *ZmSWEET4a*, *ZmSWEET4b*, *ZmSWEET13a*, *ZmSWEET13b*, and *ZmSUT4* were significantly down-regulated in the Node 8 under LN when compared with MN and HN (Figure 8). In addition, there were no differences in gene expression profiles between the MN and HN treatments.

### 2.6. Correlation Analysis

The concentrations of sucrose, fructose, and glucose in the newly expanded leaf (V8) were strongly and positively correlated with whole-plant N uptake (Figure 9). Sugar concentrations in leaves were positively correlated with the size of vascular bundles and phloem in stem nodes, but negatively correlated with their numbers. Nitrogen application resulted in an increase of the size of vascular bundle and phloem in the stem nodes, but decreased their number, and a negative relationship was observed between the vascular size and number (Figure 9). The expression levels of sugar transporters in the lower nodes were inversely related to the leaf sugar concentrations, while a positive relationship was observed in the upper Node 8 for *ZmSWEET13*. Further, sugar transporter expressions in the lower nodes were kept in line with the number of vascular bundles, while being negatively correlated with the size of vascular bundles and phloem. Such relationship was weak in the upper Node 8 (Figure 9).

## 3. Discussion

### 3.1. Structural Characteristics of Stem Nodes in Response to N Availability

Node is a junction tissue through which plant organs are connected, which contains highly developed vascular bundle systems and acts as a hub for controlling the distribution of nutrients to vegetative and reproductive organs [11]. Abundant vascular bundles were observed in either the developed stem base nodes or the upper nodes, and their structure was highly responsive to plant N availability in the present study. Recently, research found that N application significantly altered the structures of vascular bundles in the internodes at the stem base, ear-located, peduncle, and cob, in which the total number of vascular bundles of adequate N plants increased by 8–52% than in the N-deficient plants [26]. However, contrasting patterns were observed at maize vegetative phase in the present study, and low-N supply resulted in more vascular bundles and phloem in both nodes and internodes along with the stem (Figure 6; Appendix A). This discrepancy between studies might be caused by varying growth conditions, e.g., pot vs. field, sampling stages (vegetative vs. grain filling stages), or the degree of N deficiency. For instance, the maize plants grown in their zero N application treatment actually suffered moderate N stress due to relatively higher levels of soil nutrients and organic matter in the background soils, whereas more severe low-N stress was imposed in our study. Further, more vascular bundles and phloem in the low-N plants was at the expense of reduced vascular bundle size and phloem size, which was corroborated by a strong and negative correlation between vascular bundle (or phloem) size and number in nodes (Figure 6 and Figure 9). The reduction of vascular bundle size was also observed in maize when exposed to other stresses such as water deficits, while their number may be maintained or decreased depending on the timing of water deficits imposed [27]. It is likely that MN and HN plants had a similar anatomy structure of vascular bundles in the nodes or internodes under the present conditions, since there were little differences in the phenotypes of maize growth, photosynthesis, and grain yield between MN and HN treatments in most cases, as well as in sugar levels and gene expression patterns in leaves and nodes. These results suggested that maize plants can alter the structure of vascular bundles or phloem, e.g., number vs. size, to adapt to the changing environments to facilitate long-distance sugar transport.

### 3.2. Differential Responses of Sugar Partitioning to N Availability in the Lower and Upper Nodes

Transporters responsible for the allocation of mineral nutrients, e.g., phosphorus, zinc, and silicon have been identified in different cell types of vascular bundle tissues of nodes [11,12,13,28]. However, there is very limited research focused on the node characteristics associated with sugar accumulation and partitioning in plants so far, especially under abiotic stresses. In the present study, compared with the leaves, invariably greater sugar concentrations were found in the nodes across N treatments (Figure 2 and Figure 3). Greater sugar levels were found in the stem base node than in the upper nodes. Such distinction is analogous to that in the internodes in sorghum and sugarcane, where sugars tend to be accumulated in the fully elongated internodes, and to be greatest in the basal internodes and lowest in the apical internodes [29]. Further, higher monosaccharide-to-sucrose ratios were observed in all examined node tissues during vegetative phase. This was in agreement with evidence that newly developed or elongating internodes are characterized by a high ratio of hexoses to sucrose, as sucrose is rapidly cleaved to hexoses and metabolized to facilitate growth [20,30]. The higher ratio, particularly in the stem base node where nodal roots emerged, was mainly contributed by glucose in our study, which might be associated with the glucose functions in controlling root directional growth [31,32]. These results suggested that maize nodes may serve as more than a conveyor but a hub for facilitating long-distance sugar transport.

Maize adopts an apoplastic strategy of sucrose phloem loading [1,15]. It has been indicated that SWEET has a putative function to facilitate the release of sucrose into the sieve element apoplasm [16,20,21], while ZmSUT1 functions in phloem companion cells to load sucrose and also in other cell types to retrieve sucrose from the apoplasm [14,15]. In the present study, low-N supply enhanced the transcript levels of all examined sugar transporters in the three well-developed lower position Nodes 1, 2, and 3, with larger treatment differences exhibited in the lowest one. However, contrasting responses were found in the upper Node 8, where N deficiency significantly suppressed the expressions of *ZmSWEET4b*, *ZmSWEET13a*, *ZmSWEET13b*, and *ZmSUT4*, and maintained the transcript levels for other investigated transporters (Figure 8). These results indicated that the responsive patterns of sugar transporters to N availability varied with node positions or node development. A previous study has also found such disparity in the immature and mature stem internodes, namely that *ZmSUT1* is expressed in both veins and expanding storage parenchyma cells, but mainly in the veins of mature stem internodes and less in the storage parenchyma cells [15]. These results demonstrated a strong developmental impact on node function in terms of sugar transporters.

Significant correlations were established between leaf sugars and expression levels of sugar transporters in nodes. The expression levels of sugar transporters in the lower nodes were inversely related to the leaf sugar concentrations, while a positive relationship was observed in the upper Node 8 for *ZmSWEET13*. It has been found that photosynthetically fixed carbon allocation to belowground gradually decreases over the growing period in maize, indicating a strong strength of root C sink at the vegetive stages [4]. Although the root-to-shoot dry weight ratio was less affected by N treatments at V8 stage under the current conditions, the tight correlations suggested that nodes at the bottom stem might play a crucial role in carbon partitioning from source leaves to root sink.

### 3.3. Developmental Impacts on the Whole-Plant Carbon Partitioning as Affected by N Availability

Sucrose is the main form of long-distance transport of carbohydrates in most higher plants, including maize [1]. Sucrose is exported from source leaves through symplastic pathway and loaded to phloem by ZmSWEETs and ZmSUTs [14,15,16,21]. Previous study has shown that expression levels of these sucrose transporters in maize ear leaves tended to be lower than in N-sufficient plants during grain filling [6]. Therefore, it was speculated that low N conditions would affect the phloem loading, and partly limit carbon export from source leaves. In contrast, in the newly expanded leaves (the eighth leaf) during maize vegetative phase under the present conditions, low-N supply significantly upregulated most of the genes involved in hexose and sucrose transport examined in the present research (Figure 7). In parallel, a greater ^13^C amount or distribution percentages were observed in the leaf fragments below the labeling point in the low-N plants, rather than in the medium-N plants (Figure 8). The results implied that low N availability accelerated the leaf senescence and carbon export when exposed to severe N deficiency, as indicated by the dramatic reduction of N uptake and relative chlorophyll content (Table 1; Figure 1).

Leaf senescence is a developmental process needed for plant resource management [33]. In general, carbohydrate fixed during the vegetative phase is mainly utilized for cell structures which remain poorly hydrolyzed during senescence [34]. Nonetheless, the decrease in carbon during leaf senescence is around 30%, although less efficiently than N [35]. A great number of evidence has shown that leaf senescence implies the progressive reduction of the anabolic processes involved in primary carbon and N assimilation [36]. Thus, it was speculated that the newly fixed carbohydrate or labeled ^13^C-sucrose has the priority to be used for export for biomass. It is assumed that free amino acids and peptides are the predominant forms of remobilized N, which further corroborates that N remobilization is associated with carbon remobilization [33].

## 4. Materials and Methods

### 4.1. Plant Growth Conditions and Experimental Design

A greenhouse experiment was conducted at the China Agricultural University, Beijing, China, using 12.5-L pots filled with 12.8 kg air-dried soil. The experimental soil is a typical Ustochrept soil with silty loam texture. The physical and chemical characteristics were: CaCl_2_-extractable mineral N (NO_3_^−^-N and NH_4_^+^-N) 2.5 mg kg^−1^, pH (H_2_O) 7.86 (soil: water = 1:5, *w*/*v*), soil organic matter 15.5 g kg^−1^ using K_2_Cr_2_O_7_–H_2_SO_4_ digestion method, Olsen-P 16.6 mg kg^−1^, ammonium acetate extractable K 115.0 mg kg^−1^.

Three treatments were imposed in the study: low N (LN, no nitrogen fertilizer was applied as a control), medium N (MN, 0.15 g N kg^−1^ air-dried soil), and high N (HN, 0.30 g N kg^−1^ air-dried soil). A randomized complete block design with four replicates for each treatment was used. Since the LN plants suffered severe N stress, 0.03 g N kg^−1^ soil (20% of N rate in MN treatment) was top-dressed at 72 days after sowing to complete the whole life cycle. All pots received equal amount of phosphorus (P, 0.084 g kg^−1^ soil) and potassium (K, 0.12 g kg^−1^ soil). The N, P, and K nutrients were supplied with urea, monobasic potassium phosphate, and potassium sulfate, respectively, which were applied as base fertilizers and thoroughly mixed with soils before sowing. Three seeds of maize hybrid ‘Denghai 605′ (Shandong Denghai Seeds Co., Ltd., Laizhou, Shandong, China) were sown in each pot and seedlings were thinned to one per pot at the V3 (the third leaf fully expanded) stage. Plants were watered regularly during growth and soil moisture was maintained at ~70% of field capacity by a gravimetric method.

### 4.2. Growth Measurements and Plant Sampling

At 62 days after sowing (DAS), corresponding to the V8 to V9 stages (the 8th or 9th leaf fully expanded depending on the N treatments), length and maximum width of fully expanded leaves at each node were measured and used to determine the leaf area according to length × maximum width × 0.75. Chlorophyll content and net photosynthetic rate were measured on the 8th leaf. The 502 Minolta SPAD meter (Spectrum Technologies, Inc., Plainfield, IL, USA) was used to assess leaf chlorophyll content. For each measurement, five equidistant spots from the tip to the base of the leaf blade were read and averaged. Net photosynthesis was measured in plants from four individual pots at each N level between 9:00 to 11:00 h. Measurements were conducted near the middle of the leaf blade, avoiding the midrib, using a portable photosynthesis system (LI-6400XT; LI-COR, Inc., Lincoln, NE, USA) equipped with a LI-6400-02B LED light source, and conditions were set at an irradiance of 1600 µmol m^−2^ s^−1^, CO_2_ at 400 mmol m^−2^ s^−1^, flow rate at 500 mmol s^−1^, and the leaf temperature maintained at 30 ± 1 °C.

After measurements of leaf SPAD and photosynthesis, leaf and node tissues were sampled, including the 8th leaf and the corresponding node where it was located (Node 8), as well as the three lower nodes at and above the stem base (hereafter referred to as Nodes 1, 2, and 3, counted from bottom). Plants were divided into two groups. One group was designated as anatomic analysis of nodes and internodes (2 cm above nodes), which were cut from stem and immediately transferred to FAA fixative (formalin: glacial acetic acid: 50% ethanol (*v*/*v*) = 5:5:90 (*v*/*v*)). The other group destined for gene expression or carbohydrate analysis in leaves and nodes was frozen immediately in liquid N_2_ and stored at −80 °C. The remaining plant tissues were divided into shoot and root for biomass and nitrogen measurements. The concentration of nitrogen was determined by a modified Kjeldahl digestion method [37].

In addition, three replicated plants of each N treatment were grown until maturity, and maize plants were cut at the stem base and divided into roots, stalk, and kernels. Grain yield, row number per cob, kernel number per row, and one-hundred kernel weight were measured.

### 4.3. Paraffin Section and Observation

A paraffin section was prepared as described in Li et al. (2019) [27]. Node and internode tissues were first de-lignified, and followed by dehydration with a gradient alcohol series, i.e., 75% ethanol for 4 h, 85% ethanol for 2 h, 90% ethanol for 2 h, 95% ethanol for 1 h, and twice in anhydrous ethanol for 30 min. The dehydrated node and internode tissues were infiltrated with mixtures of xylene and alcohol, embedded in paraffin wax, and cut into 6 μm sections using a Leica RM 2016 microtome (Leica Shanghai Instrument Co., Ltd., Shanghai, China). The slices were finally stained with 0.5% safranin and 0.5% fast green. Microsections were scanned and then analyzed using a CaseViewer (version 2.4, 3DHISTECH Ltd., Budapest, Hungary). The size and density of vascular bundles and phloem were measured.

### 4.4. Soluble Sugar Quantification

Glucose, fructose, and sucrose were quantified using plant tissue sugar assay kits (BC2500, BC2450, and BC2460, respectively, Beijing Solarbio Science & Technology Co., Ltd., Beijing, China) according to the manufacturer’s instruction. Briefly, 100 mg fresh tissues were ground and destined for sugar extractions. Plant tissues were extracted with 10 mL 80% (*v*/*v*) ethanol in 80 °C water bath for 10 min. Extracts were centrifuged and supernatants were collected. Repeated extraction was performed for each sample and supernatants were combined. The reactions were performed with Fructose Assay Kit BC2450 or Sucrose Assay Kit BC2460 (Beijing Solarbio Science & Technology Co., Ltd., Beijing, China) according to the supplier’s protocol. The changes in the absorbance of the reaction products were monitored colorimetrically at 480 nm to quantify fructose or sucrose against a blank. For glucose assay, plant tissues were extracted with distilled water in a boiling water bath for 10 min. After cooling, the sample was centrifuged at 25 °C for 10 min, and supernatants were mixed with Glucose Assay Kit (BC2500, Beijing Solarbio Science & Technology Co., Ltd., Beijing, China), and then measured in the spectrophotometer at 505 nm following the supplier’s protocol.

### 4.5. RNA Isolation, Reverse Transcription, and Real Time PCR (qPCR)

Total RNA in the leaf and node tissues was extracted using a RNAprep pure Plant Kit (DP432, TIANGEN, Beijing, China). As described in Ning et al. (2018) [6], the quantity and quality of RNA were assessed using both a standard agarose gel electrophoretic analysis and a Nanodrop ND-1000 spectrophotometer. Afterwards, the isolated RNA sample was used as template for complementary DNA (cDNA) synthesis using PrimeScript™ RT reagent Kit with gDNA Eraser (Takara, Biomedical Technology (Beijing) Co., Ltd., Beijing, China). For qPCR, SYBRTM Green Master Mix (Takara, Biomedical Technology (Beijing) Co., Ltd., Beijing, China ) was used in the reaction mixture according to the manufacturer’s instructions. Reactions were performed in 96-well plates on an ABI 7500 real-time PCR system with universal cycling conditions (95 °C for 2 min, 40 cycles of 95 °C for 30 s, and 60 °C for 30 s). The qPrimerDB-qPCR Primer Database was used to obtain primers, which are listed in the Appendix A. *ZmTublin* was used as a reference gene. Genes putatively involved in sucrose transport were investigated, including *ZmSWEET3a* (*GRMZM2G179679*), *ZmSWEET4a* (*GRMZM2G000812*), *ZmSWEET4b* (*GRMZM2G144581*), *ZmSWEET13a* (*GRMZM2G173669*), *ZmSWEET13b* (*GRMZM2G021706*), *ZmSWEET13c* (*GRMZM2G179349*), *ZmSUT1* (*GRMZM2G034302*), *ZmSUT2* (*GRMZM2G307561*), and *ZmSUT4* (*GRMZM2G145107*). Relative expression values were calculated according to the 2^−ΔΔCt^ method, and were expressed as fold changes relative to values of low N supply (LN).

### 4.6. ^13^C-Sucrose Export Assay

At 62 DAS, the 8th leaves of four maize plants were selected from each treatment and were labeled with ^13^C-labeled sucrose (99% atom% ^13^C, IR-25551, ISOREAG, Shanghai, China) as described by Slewinski et al. (2009) [14] with a few modifications, i.e., ^13^C was used instead of ^14^C. Briefly, the adaxial surface was gently abraded (~0.25 cm^2^) with a find grain sandpaper to remove the cuticle and epidermis at 10 cm below the leaf tip. A 20 μL solution containing 17.2 μmol L^−1^ of ^13^C-labeled sucrose was applied to the abraded site and covered with Parafilm to prevent dehydration for 1.5 h. After labeling, five leaf fragments were sampled from the labeled leaves, including ^13^C-application section (0 cm) and four sections below it every 2 cm (0–2, 2–4, 4–6, and 6–8 cm). The ^13^C labeling leaf fragment was washed thoroughly with deionized water to remove the surface adhered ^13^C-sucrose. All ^13^C labeling work was conducted between 10:00 to 12:00 in the morning. After sampling, the leaf fragments were immediately heated at 105 °C for 30 min and dried at 60 °C to a constant weight. The dried tissues were ground using a ball mill (MM400, Retsch, Haan, Germany) for ^12^C/^13^C analysis, using an isotope ratio mass spectrometer (Delta plus xp, Thermo Fisher SCIENTIFIC, Waltham, MA, USA). The content of ^13^C incorporated into the leaf fragments were calculated based on the following equations according to Zhao et al. (2019) [38].
^13^C content = DM × C% × (Atm%_(Sample)_ − Atm%_(control)_) × 100(1)
where ^13^C content (μg) is ^13^C content in each leaf fragment, DW (g) represents the biomass of the samples, C% refers to the percentage of C mass in the DW present, Atm% is the ratio of ^13^C isotope atomic number to total atomic number of C (sum of ^13^C and ^12^C) in the 8th leaf of labeled and unlabeled control plants.
(2)Total C13 content=∑n=15C13 content of Sample(n)
(3)C13 distribution (%)=C13 content (sample)Total C13 content×100

### 4.7. Statistical Analyses

Data were subjected to homogeneity variance analysis, and followed by one-way analysis of variance (ANOVA) using SPSS software (v23.0.0.0, IBM Corp., Armonk, NY, USA) in which nitrogen rate was treated as the fixed effect and replication as the random effect. The Fisher’s least significant difference (LSD) was used to compare treatment differences at a *p* < 0.05 level of probability. The observation data of vascular bundles between LN and MN treatments were tested by the two-tailed *t* test with SPSS software. Further, the Pearson correlation analysis was performed using SPSS software and the *p* values were not adjusted.

## 5. Conclusions

The present study demonstrated that stem nodes may play an important role in facilitating long-distance sugar transport in maize. This was revealed by the differential responsive patterns of sugar accumulation, sugar transporters, and anatomical structure in N-deficient and N-sufficient plants. Low-N supply resulted in more vascular bundles and phloem in both nodes and internodes along with the stem, which was at the expense of reduced vascular bundle size and phloem size. These results indicated that maize node structure can be optimized to facilitate long-distance sugar transport to adapt to changing environments, such as in nutrient shortage. Regarding the sugar transporters’ putative function in stem nodes, their responsive patterns to N availability varied with node positions or node development. Low-N supply up-regulated their expressions in the bottom nodes but suppressed them in the upper ones, which might be associated with the strong root C demand during vegetive phase. Further research is needed to elucidate the differential impacts of N availability on sugar transport in nodes located at different stem positions, and their associations with the whole-plant carbon partitioning in maize.

## Figures and Tables

**Figure 1 ijms-23-04389-f001:**
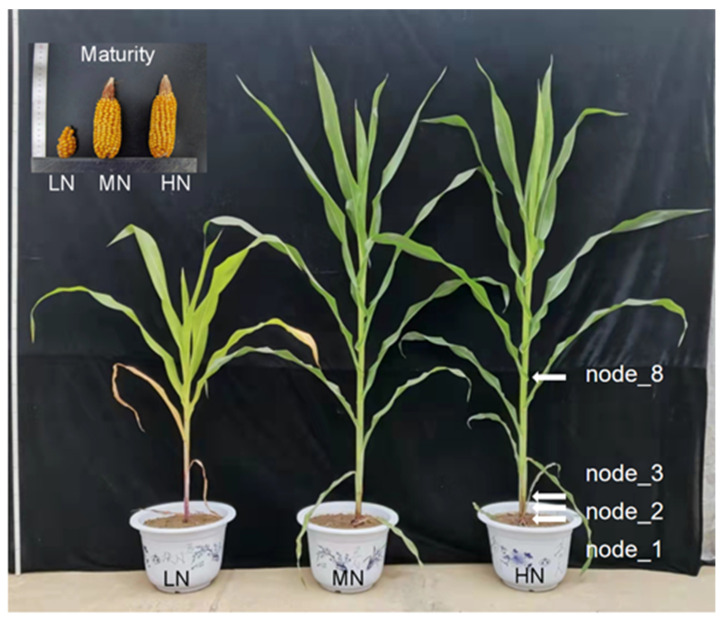
Phenotypes of maize plant at 62 days after sowing and ears at maturity under low to high nitrogen supply. LN, no nitrogen application; MN, 0.15 g N kg^−1^; HN, 0.30 g N kg^−1^; Arrows indicate the stem node position.

**Figure 2 ijms-23-04389-f002:**
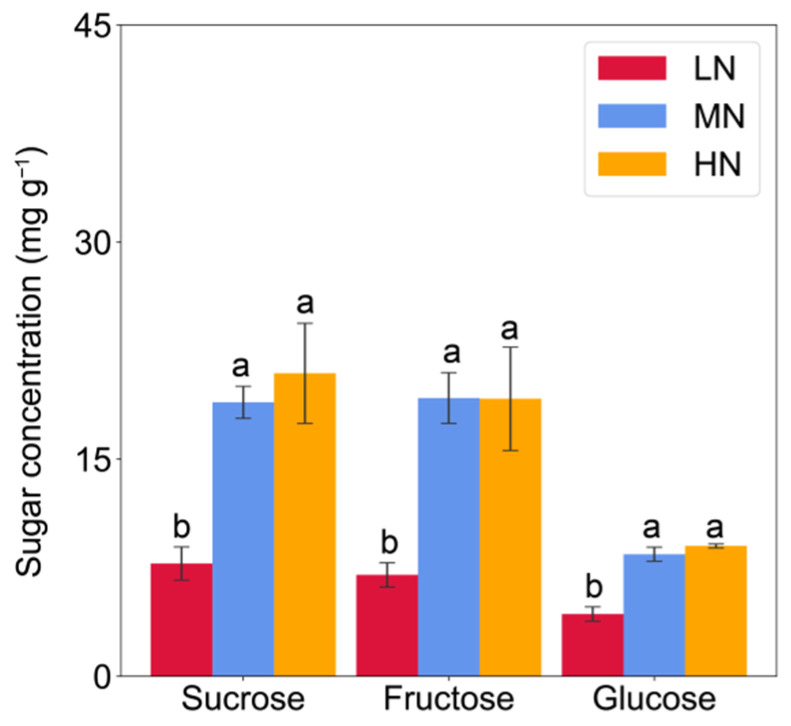
Sugar concentrations in the leaf (the 8th fully expanded) of maize plants under low, medium, and high nitrogen supply at 62 days after sowing. Vertical bars indicate the standard error of the mean (*n* = 4). Different letters above the columns represent significant differences within each sugar between nitrogen treatments (*p* < 0.05). LN, no nitrogen application; MN, 0.15 g N kg^−1^; HN, 0.30 g N kg^−1^.

**Figure 3 ijms-23-04389-f003:**
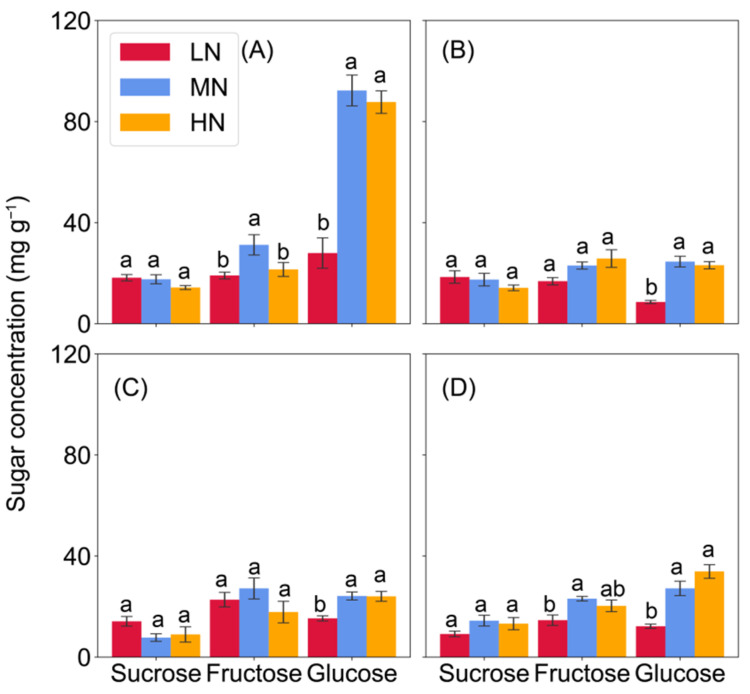
Sugar concentrations in the nodes of maize plants under low, medium, and high nitrogen supply at 62 days after sowing. (**A**) Stem base node (node 1); (**B**,**C**) two nodes above the stem base node (Nodes 2 and 3, respectively); (**D**), Node 8 where the 8th leaf is located. Vertical bars indicate the standard error of the mean (*n* = 4). Different letters above the columns in each panel represent significant differences between nitrogen treatments for each sugar (*p* < 0.05). LN, no nitrogen application; MN, 0.15 g N kg^−1^; HN, 0.30 g N kg^−1^.

**Figure 4 ijms-23-04389-f004:**
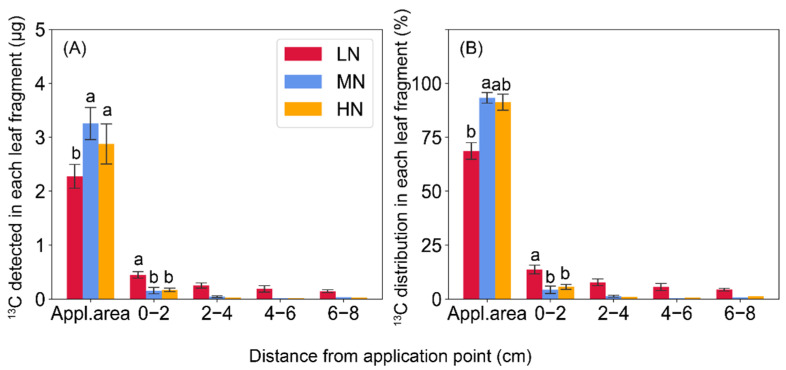
The absolute amount (**A**) and relative distribution (**B**) of ^13^C in the 8th fully expanded leaf following foliar application of ^13^C labeled sucrose at the leaf tip. Columns with no letter in common are significantly different between nitrogen treatments within each leaf fragment (*p* < 0.05). Vertical bars indicate the standard error of the mean (*n* = 4). Error bars are not shown for leaf fragments where ^13^C was only detected in one or two of the four replicates. Appl. area means the application area of ^13^C-sucrose labelled near the leaf tip.

**Figure 5 ijms-23-04389-f005:**
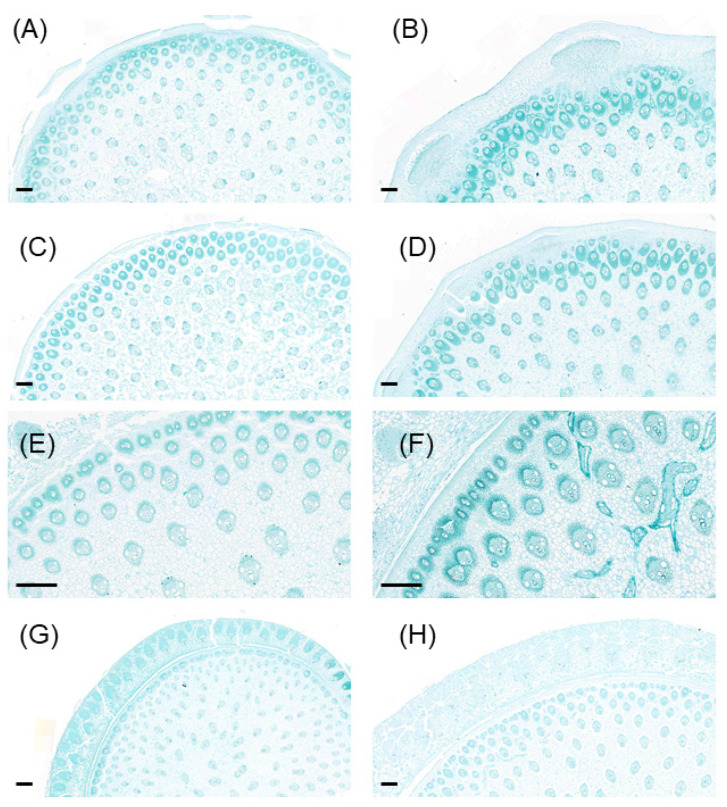
Transverse sections show the stem base node (Node 1, (**A**,**B**)), two nodes above stem base ((**C**,**D**) for Node 2, and (**E**,**F**) for Node 3, respectively), and Node 8 (**G**,**H**) in maize plants under low (**A**,**C**,**E**,**G**) and medium N (**B**,**D**,**F**,**H**) availability at 62 days after sowing. Scale bars represent 500 μm.

**Figure 6 ijms-23-04389-f006:**
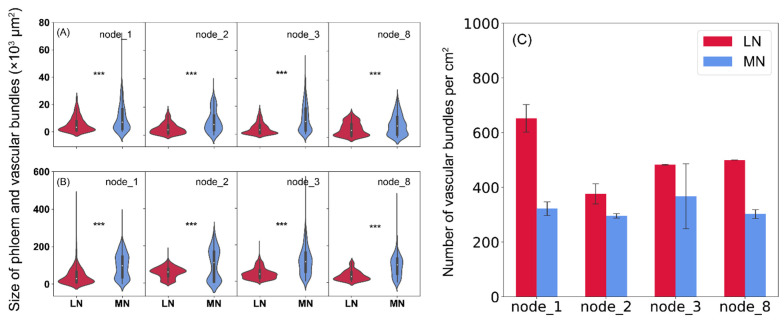
The size of phloem (**A**) and vascular bundles (**B**), and number of vascular bundles per unit area (cm^2^) (**C**) in the stem base node (node_1), two nodes above stem base (node_2 and node_3), and the node_8 of maize plants at 62 days after sowing under low and medium N supply. The solid line and dash line within each box represent the median and mean values of all data in panels (**A**) and (**B**). The top and bottom edges represent the 75 and 25 percentiles, and the top and bottom bars represent the 95 and 5 percentiles of all data, respectively. ***, *p* < 0.001.

**Figure 7 ijms-23-04389-f007:**
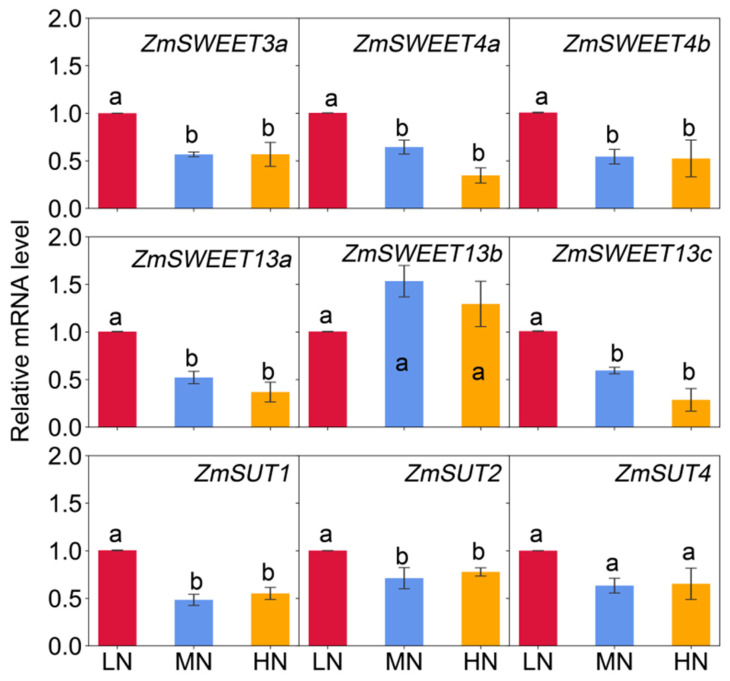
The relative expression level of genes involved in sugar transport in leaves at 62 days after sowing. Columns with no letter in common are significantly different between nitrogen treatments in each panel (*p* < 0.05). Vertical bars indicate the standard error of the mean (*n* = 4).

**Figure 8 ijms-23-04389-f008:**
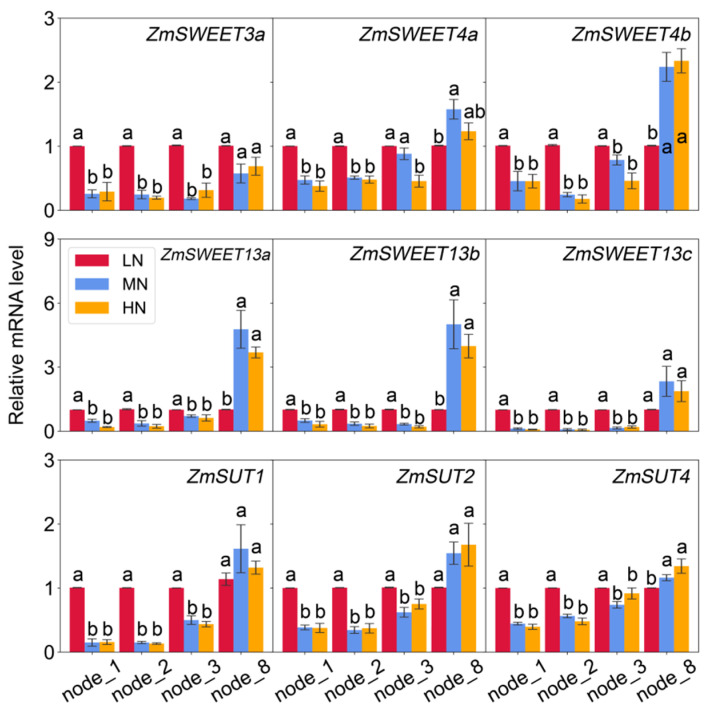
The relative expression level of genes putatively involved in sugar transport in nodes at 62 days after sowing. Columns with no letter in common are significantly different between nitrogen treatments in each panel (*p* < 0.05). Vertical bars indicate the standard error of the mean (*n* = 4).

**Figure 9 ijms-23-04389-f009:**
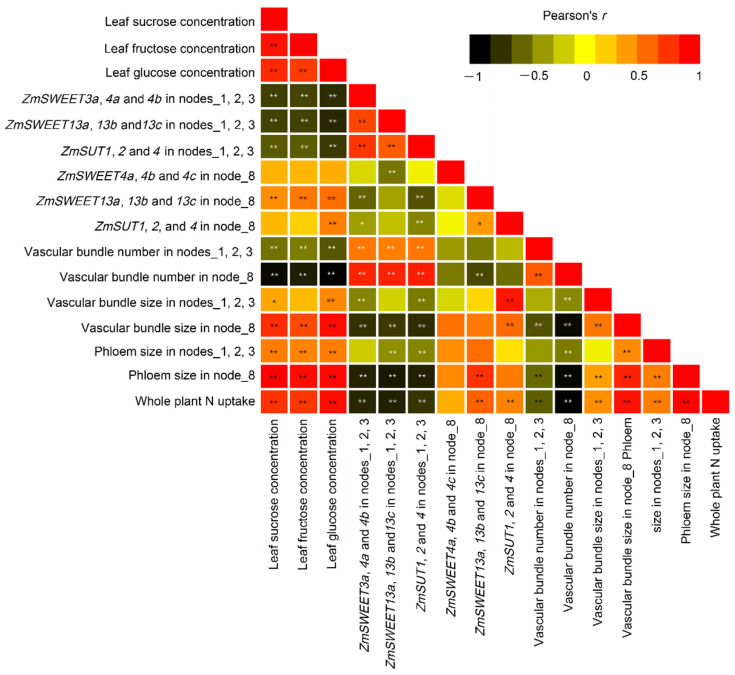
Correlation analysis of sugar concentrations, expression levels of sugar transporters, vascular bundle morphology, and total N uptake in maize plants. *, *p* < 0.05; **, *p* < 0.01.

**Table 1 ijms-23-04389-t001:** Effects of nitrogen (N) availability on maize shoot and root dry matter (DM), N uptake, leaf SPAD readings, leaf area, and photosynthesis (P_n_) at 62 days after sowing (DAS) ^1^.

Items	LN	MN	HN
Shoot DM (g plant^−1^)	16.23 ± 1.17 b	43.41 ± 2.48 a	45.99 ± 1.96 a
Root DM (g plant^−1^)	5.58 ± 0.64 b	15.82 ± 1.20 a	16.18 ± 1.25 a
R/S ratio	0.35 ± 0.04 a	0.36 ± 0.02 a	0.36 ± 0.04 a
Total N uptake (g plant^−1^)	0.20 ± 0.02 c	1.05 ± 0.05 b	1.24 ± 0.03 a
Total leaf area (cm^2^)	2553.6 ± 134.0 b	4394.8 ± 132.1 a	4346.3 ± 64.9 a
SPAD	22.30 ± 0.69 b	48.05 ± 0.55 a	49.05 ± 1.23 a
P_n_ (μmol CO_2_ m^−2^ s^−1^)	8.00 ± 1.08 b	23.24 ± 1.14 a	21.80 ± 2.43 a

^1^ Data are mean ± standard error (*n* = 4). Means with no letter in common indicate significant differences between N treatment for each parameter (*p* < 0.05). LN, no nitrogen application; MN, 0.15 g N kg^−1^; HN, 0.30 g N kg^−1^.

## Data Availability

Not applicable.

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
