# Peer review of "Characterization of Stem Nodes Associated with Carbon Partitioning in Maize in Response to Nitrogen Availability"

_ijms, 2022, doi:10.3390/ijms23084389_

Round 1

Reviewer 1 Report

In this manuscript, Zhao et al explores the nitrogen effect on the carbon partitioning through stem nodes in maize. They found that low nitrogen (LN) results in much lower sugar content in leaves and most of the nodes. They found the LN-treated plants have more but smaller vascular bundles. In addition, they used qRT-PCR to show the LN treatment increases the sugar transporters’ gene expression in the bottom stem node, but decreases sugar transporters’ gene expression in the upper node 8. The author concluded the stem nodes in maize may play an important role in the long-distance sugar transportation.

I appreciate that the author did the sugar content experiment in multiple nodes. And I could the manuscript will be interesting to many reads. I have some concerns and suggestions about the manuscript that are listed below.

  1. In line “Nitrogen (N) is one of the critical mineral nutrients affecting plant growth”. I don’t think nitrogen is considered as “mineral nutrients”.
  2. For figure 2 and figure 3, please make the bars (three colors) next to each other, like Figure 4. Do not stack them as stacking makes plot very hard to compare.
  3. In section 2.4, the authors compared the stem anatomy structure between LN and MN. How about HN? Is HN and MN very similar? If so, please provide the data. The figure could be added as a supplemental material. I think this information will be very interesting to readers.
  4. For ZmSWEETs and ZmSUTs genes, please add the gene IDs in the main text. I know the gene IDs are in the supplementary table2, but adding them in the main text will be helpful to readers.
  5. There are many members of SWEET and SUT genes. Please provide references or evidence that these six ZmSWEETs and three ZmSUTs are the major players in sugar transportation in maize.
  6. In line 184, please separate Figure 7 and 8. The sentence is confusing. Maybe split into two sentences as Figure 7 is leaf and Figure 8 is node.
  7. Please enhance the Figure 9’s quality.
  8. For the Figure 9, how is correlation calculated? In SPSS, R or some other software? Are p-values adjusted for multiple testing? If so, how? Please add a paragraph in the method section about this analysis.
  9. In figure9, some abbreviations are not clear. For example, what does Sweet4 represent? There are Sweet4a and Sweet4b. The same goes to “Sut”. Also, I assume “VB No.” means Vascular Bundle Number. Please write the full name in figure legend.
  10. Line 229, please explain the differences in detail. Why the current study differs from previous one? For example, what is the sample stage is the ref25? What genotype does ref25 used?
  11. Line 299, I'm not sure about the link with senescence. The sampling 8th leaf is not senescencing, right? I can understand under LN the carbon is exported faster. But I'm not clear about the link with senescence.
  12. Line 316, when is the nitrogen given? At the beginning of the experiment (V3 stage) or later?
  13. I’m not clear about the statistical analysis the author did. According to section 4.7, I assume the author fit an LMM model in SPSS. Please write down the model formula in the method. Also has the p-value been muti-testing adjusted? If so, which method was used? Please explain the LMM model and the statistical analysis in detail.
  14. Some sentences in the manuscript are too long. Please split them into two or three shorter sentences. For example, in section 5- conclusion, the sentences are very long and are hard to follow.

Author Response

Response to Reviewer 1 Comments

In this manuscript, Zhao et al explores the nitrogen effect on the carbon partitioning through stem nodes in maize. They found that low nitrogen (LN) results in much lower sugar content in leaves and most of the nodes. They found the LN-treated plants have more but smaller vascular bundles. In addition, they used qRT-PCR to show the LN treatment increases the sugar transporters’ gene expression in the bottom stem node, but decreases sugar transporters’ gene expression in the upper node 8. The author concluded the stem nodes in maize may play an important role in the long-distance sugar transportation.

I appreciate that the author did the sugar content experiment in multiple nodes. And I could the manuscript will be interesting to many reads. I have some concerns and suggestions about the manuscript that are listed below.

Point 1: In line “Nitrogen (N) is one of the critical mineral nutrients affecting plant growth”. I don’t think nitrogen is considered as “mineral nutrients”.

Response 1: We have modified the sentence to “Nitrogen (N) is one of the critical nutrients affecting plant growth” in the revised version.

Point 2: For figure 2 and figure 3, please make the bars (three colors) next to each other, like Figure 4. Do not stack them as stacking makes plot very hard to compare.

Response 2: We have redrawn the figures 2 and 3 according to the reviewer’s comments. In the previous version, one of the reasons that we made the stacked bar plots was to compare the differences in total sugars between N treatments. To this end, we added a supplementary table containing such comparisons in the supporting materials in the revised version.

Point 3: In section 2.4, the authors compared the stem anatomy structure between LN and MN. How about HN? Is HN and MN very similar? If so, please provide the data. The figure could be added as a supplemental material. I think this information will be very interesting to readers.

Response 3: The anatomic analysis of HN plants was not performed in the present study, and such limitations have been discussed in the subsection 3.1 in the revised version. It is likely that MN and HN plants had similar anatomy structure of vascular bundles in the nodes or internodes under the present conditions, since there were little differences in phenotypes of maize growth, photosynthesis and grain yield between MN and HN treatments in most cases, as well as in sugar levels and gene expression patterns in leaves and nodes.

Point 4: For ZmSWEETs and ZmSUTs genes, please add the gene IDs in the main text. I know the gene IDs are in the supplementary table2, but adding them in the main text will be helpful to readers.

Response 4: We have added the gene IDs in the subsection 4.5 of Materials and Methods in the revised version.

Point 5: There are many members of SWEET and SUT genes. Please provide references or evidence that these six ZmSWEETs and three ZmSUTs are the major players in sugar transportation in maize.

Response 5: We have added references in the main text according to the reviewer comments. Please see the subsection 2.5 Transcript abundance of genes involving in sugar transport in leaves and nodes in the revised version.

Point 6: In line 184, please separate Figure 7 and 8. The sentence is confusing. Maybe split into two sentences as Figure 7 is leaf and Figure 8 is node.

Response 6: We have modified the sentence, and Figures 7 and 8 were cited separately in the revised version.

Point 7: Please enhance the Figure 9’s quality.

Response 7: We have improved the image resolution of Figure 9.

Point 8: For the Figure 9, how is correlation calculated? In SPSS, R or some other software? Are p-values adjusted for multiple testing? If so, how? Please add a paragraph in the method section about this analysis.

Response 8: We have added more details for the correlation analysis in the revised version. Please see the subsection 4.7 Statistical analyses.

Point 9: In figure9, some abbreviations are not clear. For example, what does Sweet4 represent? There are Sweet4a and Sweet4b. The same goes to “Sut”. Also, I assume “VB No.” means Vascular Bundle Number. Please write the full name in figure legend.

Response 9: Figure 9 has been updated in the revised version, and some abbreviations have been replaced by their full names.

Point 10: Line 229, please explain the differences in detail. Why the current study differs from previous one? For example, what is the sample stage is the ref25? What genotype does ref25 used?

Response 10: We have added more discussion in the corresponding sentences as following, “This discrepancy between studies might be caused by varying growth conditions, e.g. pot vs. field, sampling stages (vegetative vs. grain filling stages) or degrees of N deficiency. For instance, the maize plants grown in their zero N application treatment were actually suffered moderate N stress due to relatively higher levels of soil nutrients and organic matter in the background soils, whereas more severe low-N stress was imposed in our study.”

Point 11: Line 299, I'm not sure about the link with senescence. The sampling 8th leaf is not senescencing, right? I can understand under LN the carbon is exported faster. But I'm not clear about the link with senescence.

Response 11: Compared to the sufficient N plants, SPAD readings in the 8th leaves of low-N plants decreased by 54-55% (Table 1; Figure 1). Therefore, the leaf senescence was accelerated by low-N stress. We have modified the sentences as follow “The results implied that low N availability accelerated the leaf senescence and carbon export when exposed to severe N deficiency as indicated by the dramatic reduction of N uptake and relative chlorophyll content (Table 1; Figure 1)”.

In general, carbohydrate fixed during the vegetative phase is mainly utilized for cell structures which remain poorly hydrolyzed during senescence. Nonetheless, the decrease in carbon during leaf senescence is around 30%, although less efficiently than N. A number of evidence have shown that leaf senescence implies the progressive reduction of the anabolic processes involved in primary carbon and N assimilation. Thus, it was speculated that the newly fixed carbohydrate or labeled 13C-sucrose has the priority to be used for export for biomass. This information has been discussed at the end of subsection 3.3.

Point 12: Line 316, when is the nitrogen given? At the beginning of the experiment (V3 stage) or later?

Response 12: Nitrogen was applied in the form of urea before sowing and thoroughly mixed with soils as indicated in the 2nd paragraph of subsection 4.1.

Point 13: I’m not clear about the statistical analysis the author did. According to section 4.7, I assume the author fit an LMM model in SPSS. Please write down the model formula in the method. Also has the p-value been muti-testing adjusted? If so, which method was used? Please explain the LMM model and the statistical analysis in detail.

Response 13: We have modified the descriptions of Statistical analyses in the revised version as follows, “Data were subjected to homogeneity variance analysis, and followed by one-way analysis of variance (ANOVA) using SPSS software (version 23.0.0.0, IBM Corp, America) in which nitrogen rate was treated as the fixed effect and replication as the random effect. The Fisher’s least significant difference (LSD) was used to determine treatment differences at a p < 0.05 level of probability. The observation data of vascular bundles between LN and MN treatments were tested by two-tailed t test with SPSS software. Further, the Pearson correlation analysis was performed using SPSS software and the p values were not adjusted.”

Point 14: Some sentences in the manuscript are too long. Please split them into two or three shorter sentences. For example, in section 5- conclusion, the sentences are very long and are hard to follow.

Response 14: We have carefully gone through the main text, and reworded some descriptions in the revised version. Edits made to the text are marked with tracking changes.

Reviewer 2 Report

The manuscript by Zhao et al. presents the maize response to nitrogen availability characterized by the analysis of plant growth and anatomy of nodes, sugar concentration and analysis of the expression of genes related to sugar transporters (SUT, SWEET). The Authors properly introduced into their research area, performed experiments and provide conclusions. The research objective is in the scope of IJMS. I have only minor comments/questions:

Net assimilation is relatively low (22-23 μmol CO2 m-2 s-1 for HN and MN variants), taking into account the tested plant species (C4), development stage (V8) and conditions (irradiance 1800 μmol m-2 s-1, leaf temp. 30 C). Please explain the possible reason for this observation. How many plants (per each variant) were used for measurements? I am also curious what the results of the chlorophyll fluorescence measurements looked like. The equipment used (LI-COR) allows the measurement of many parameters of chlorophyll fluorescence, including these at the light, simultaneously with the assimilation measurements.

Author Response

Response to Reviewer 2 Comments

The manuscript by Zhao et al. presents the maize response to nitrogen availability characterized by the analysis of plant growth and anatomy of nodes, sugar concentration and analysis of the expression of genes related to sugar transporters (SUT, SWEET). The Authors properly introduced into their research area, performed experiments and provide conclusions. The research objective is in the scope of IJMS. I have only minor comments/questions:

Point1: Net assimilation is relatively low (22-23 μmol CO2 m-2 s-1 for HN and MN variants), taking into account the tested plant species (C4), development stage (V8) and conditions (irradiance 1800 μmol m-2 s-1, leaf temp. 30 °C). Please explain the possible reason for this observation. How many plants (per each variant) were used for measurements? I am also curious what the results of the chlorophyll fluorescence measurements looked like. The equipment used (LI-COR) allows the measurement of many parameters of chlorophyll fluorescence, including these at the light, simultaneously with the assimilation measurements.

Response 1:

(1) The irradiance was set to 1600 μmol m-2 s-1 in the present study, and we have corrected it in the revised version. We found the photosynthetic rate of ear leaf in other study was also lower, which was conducted in the same green house with same instrument and irradiance (Please see the figure attached below). Generally, the light intensity in greenhouse is relatively lower than that in the field. This might be the main reason why the overall photosynthesis was lower than expected.

(2) Four maize plants from four different pots were used for measurements in each N treatment. We have added this information in the revised version.

(3) The reviewer suggested an interesting point about the responses of chlorophyll fluorescence. We do agree with the reviewer that fluorescence parameters are closely associated with carbon assimilation. However, photosynthesis was measured using a LI-COR instrument equipped with a LI-6400-02B LED light source, instead of an integrated fluorescence chamber head, therefore the chlorophyll fluorescence parameters were not collected and analyzed in the current study.

Figure for review only (Adapted from Yang, 2017, Ph.D Dissertation). Mazie hybrid “Zhengdan 958” was grown in the same green house with our study, and photosynthesis was measured using the same LI-COR instrument at same photo flux density of 1600 μmol m-2 s-1.

Yang L, 2017, Physiological mechanism of leaf senescence and nitrogen remobilization in maize (Zea mays L.) as affected by source and sink manipulation. Ph.D Dissertation, China Agricultural University, Beijing, China (In Chinese with English Abstract).

Reviewer 3 Report

The work is undoubtedly interesting and worth publishing, the research was carried out using a large number of diverse microscopic, anatomical and genetic methods. Very important studies have been performed on the expression of sugar transporters in individual parts of the plant. However, the authors too weakly emphasized in the introduction that the detection of the role of stem nodes in facilitating the transport of sugar is really novel. This hypothesis should be more closely related to this finding.

Descriptions of abbreviations are missing in the titles of figures, they should be attached

Figs. 4 The difference between the detected 13C (Fig.4A) and its distribution (Fig.4B) is unclear – this needs to be clarified.

The soil used for the research is described too briefly and in general. The granulometric composition of this soil and the content of humic acids should be specified.

Without providing a method for determining the organic matter content, it is not clear what fraction of soil is meant.

Total The C and N content must be determined, specifying the methods used and the C:N ratio calculated.

The content of components in the soil as well as the amount of NPK introduced must be given in the same units per unit of soil mass (kg or g) – the amount of NPK introduced into the pots (g per pot) per unit of soil dry matter should be converted.

The use of urea as a source of nitrogen is not good, because this compound undergoes very rapid microbiological degradation, which entails losses of N from the soil. The content of total nitrogen and its nitrate and ammonium forms should be checked several times during the 62-day experiment.

Author Response

Response to Reviewer 3 Comments

Point 1: The work is undoubtedly interesting and worth publishing, the research was carried out using a large number of diverse microscopic, anatomical and genetic methods. Very important studies have been performed on the expression of sugar transporters in individual parts of the plant. However, the authors too weakly emphasized in the introduction that the detection of the role of stem nodes in facilitating the transport of sugar is really novel. This hypothesis should be more closely related to this finding.

Response 1: We appreciated the reviewer’s comments, and made some edits to improve the novelty of our study. Please see the revised introduction.

Point 2: Descriptions of abbreviations are missing in the titles of figures; they should be attached Figs. 4 The difference between the detected 13C (Fig.4A) and its distribution (Fig.4B) is unclear – this needs to be clarified.

Response 2: We have added the explanation of abbreviations in the figure caption in the revised version. The differences between Fig. 4A and Fig. 4B were also distinguished as they present the absolute amount of 13C and relatively distribution in each leaf fragment, respectively. Please see the revised figure caption.

Point 3: The soil used for the research is described too briefly and in general. The granulometric composition of this soil and the content of humic acids should be specified.

Response 3: We have added the soil texture information in the revised version. The content of humic acids was not determined, but can be referred to the soil organic matter.

Point 4: Without providing a method for determining the organic matter content, it is not clear what fraction of soil is meant.

Response 4: We have added the methods of soil organic matter determination, as well as other soil properties in the revised version.

Point 5: Total C and N content must be determined, specifying the methods used and the C:N ratio calculated.

Response 5: We just analyzed the basic soil properties related to soil fertility, including pH, soil organic matter, and available nutrients, however, total C and total N and their ratios were not measured. For the present study focused on N impacts, soil N availability is crucial for the validation of results obtained, especially for the low-N treatment. According the mineral N content (NO3--N and NH4+-N) 2.5 mg kg-1, the experimental soil can be considered to be a N-deficient soil.

Point 6: The content of components in the soil as well as the amount of NPK introduced must be given in the same units per unit of soil mass (kg or g) – the amount of NPK introduced into the pots (g per pot) per unit of soil dry matter should be converted.

Response 6: The inputs of P and K were converted and expressed as per unit of soil mass in the revised version.

Point 7: The use of urea as a source of nitrogen is not good, because this compound undergoes very rapid microbiological degradation, which entails losses of N from the soil. The content of total nitrogen and its nitrate and ammonium forms should be checked several times during the 62-day experiment.

Response 7: In order to observe a N-deficient phenotype of maize plants in the present pot experiment, we used a soil containing lower available N (Nmin, 2.5 mg kg-1) and soil organic C (9.0 g kg-1). Such soil properties might be favor of microbial growth to obtain both C and N sources from urea. The reason we choose urea as a N source is that it is widely applied in farmers’ practice. Although N utilization efficiency is generally higher in pot experiment due to the confined root growth spaces than in the field, we do agree with the reviewer that a proportion of N applied was lost. A continuous soil sampling would take N nutrient out of the pots. Alternatively, a destructive sampling might be help but needs several-folds more pots set up at the beginning. Therefore, we did not monitor the dynamics of soil nitrate and ammonium during the 62-day cultivation. However, according to the plant traits of growth, N uptake, and grain yield, it is valid that LN treatment exerted severe low-N stress to plants, and MN and HN soils had higher N availability under the present conditions.